# Effect of Near-Freezing Storage Combined with High-Voltage Electric Fields on the Freshness of Large Yellow Croaker

**DOI:** 10.3390/foods13182877

**Published:** 2024-09-11

**Authors:** Hui Zhang, Qizhang Yang, Songyi Lin, Zhaobin Yan, Xuancheng Wu, Wenqiang Wei, Guibing Pang

**Affiliations:** 1School of Mechanical Engineering and Automation, Dalian Polytechnic University, Dalian 116034, China; zh1226419340@163.com (H.Z.); sxyangqizhang@126.com (Q.Y.); 17866511244@163.com (X.W.); 15135719002@163.com (W.W.); 2SKL of Marine Food Processing & Safety Control, National Engineering Research Center of Seafood, School of Food Science and Technology, Dalian Polytechnic University, Dalian 116034, China; linsongyi730@163.com; 3Zhejiang TOPSUN Logistic Control Co., Ltd., Yuhuan 317600, China; 4School of Food Science and Technology, Dalian Polytechnic University, Dalian 116034, China; yanzhaobin2022@163.com

**Keywords:** near-freezing storage, high-voltage electric field (HVEF), large yellow croaker, total viable counts, shelf life

## Abstract

Seafood is highly perishable after being caught, making effective preservation technology essential. A few studies have explored the mechanisms of near-freezing storage combined with high-voltage electric fields for seafood preservation. This study uses near-freezing storage at −1 °C in conjunction with three high-voltage electric fields (5 kV/m, 8 kV/m, and 16 kV/m) to store large yellow croakers for 21 days and assesses their quality through sensory evaluation, pH values, malondialdehyde, total volatile basic nitrogen, and total viable counts. The results indicate that high-voltage electric fields effectively inhibit endogenous enzyme activity and microbial growth while reducing lipid oxidation in large yellow croakers. The preservation effect is optimal at an electric field strength of 16 kV/m, extending their shelf life by 9 days. These findings offer valuable theoretical and data-driven insights for applying near-freezing storage and electric field preservation technology in cross-regional fish transportation.

## 1. Introduction

The large yellow croaker is a valuable marine resource with considerable economic importance. Its meat is rich in protein, unsaturated fatty acids, and various trace elements, providing a well-balanced source of nutrition when consumed in moderation [1,2]. After being caught, large yellow croaker is typically marketed in chilled form. During this storage period, increased enzyme activity and rapid microbial growth can deteriorate the fish’s quality and cause spoilage [3,4]. Consequently, to preserve the freshness of large yellow croaker and extend its shelf life [5,6], researchers are investigating new auxiliary preservation and storage technologies, including ultra-cold, biological, and modified-atmosphere techniques.

Research on advanced ultra-cold technology found that the yellowness values and texture of large yellow croakers increased by 20% and 15%, respectively, when the fish was ultra-cooled with liquid nitrogen at −60 °C and then stored at −18 °C for 6 months compared to being ultra-cooled at −40 °C and then stored at −18 °C for the same duration [7]. Bio-preservation technology commonly utilizes chitosan, clove essential oil, and other preparations to extend the shelf life and enhance the color of fish meat. Using chitosan and lysozyme as biological preservatives for large yellow croakers demonstrated a strong antibacterial effect, reducing the number of microorganisms by 0.5–1 log during the storage period [8]. For tuna slices, tannin treatment combined with a modified atmosphere effectively minimized discoloration and lipid oxidation [9]. Moreover, storing flounder fillets at 10 °C in a modified atmosphere with a gas composition of CO_2_ 60%/O_2_ 30%/N_2_ 10% helped to maintain low total volatile basic nitrogen levels and exhibited antibacterial properties [10]. However, due to technical limitations, ultra-cold, biological, and controlled-atmosphere preservation technologies have certain drawbacks. For example, ultra-cold preservation cannot ensure freshness throughout the entire supply chain and is energy-intensive. The selection of biological preservatives must be carefully matched to the specific characteristics of the seafood [11], and there may be consumer concerns regarding food safety, potentially affecting their willingness to buy these products. Additionally, selecting and proportioning gases for controlled atmosphere preservation are complex [12].

The electric field is an emerging physical preservation technology that offers environmental benefits, is low cost, and is low in energy consumption due to its non-thermal effects [13]. Researchers have explored its potential for preserving fruits and vegetables [14,15,16], meat [17], and aquatic products [18]. A study found that a low-voltage electric field reduced enzyme activity and respiration rates in mushrooms, extending their shelf life by 2 days under refrigerated conditions (4 °C) [19]. A low-voltage electric field (1.20 kV/m) reduced protein and lipid oxidation in pigeon meat during refrigeration (3 °C) [20]. For large yellow croakers, combining a low-voltage electric field with micro-freezing (−3 °C) effectively maintained sensory quality, inhibited bacterial growth, preserved myofibrillar tissue, and extended shelf life by 3 to 6 days [21]. However, there is still potential for further development of electric field preservation technology specifically for aquatic products. Research has shown that high-voltage electric fields can lower the freezing point of fish by 1–2 °C; that is, the freezing point of fish decreases to −2 °C to −4 °C under the influence of high-voltage electric fields. Moreover, increasing the electric field strength can affect cell membrane permeability [22]. Thus, further study is needed on the mechanisms of combining near-freezing storage (−1 °C) with high-voltage electric fields to preserve large yellow croakers.

This paper aimed to investigate the combined effects of high-voltage electric fields and near-freezing storage on the comprehensive quality of large yellow croaker. Near-freezing storage (−1 °C) was used as the control group, and the effects of high-voltage electric fields (5 kV/m, 8 kV/m, and 16 kV/m) combined with near-freezing storage (−1 °C) on the comprehensive quality of large yellow croaker were analyzed, using indicators including sensory evaluation value, pH value, malondialdehyde, total volatile basic nitrogen, and total viable counts. This study aimed to provide theoretical and data-driven references for the application of near-freezing storage and electric field preservation processes in the cross-regional transportation of fish. 

## 2. Design of Experiments

### 2.1. Materials and Main Reagents

Fresh large yellow croakers caught within one hour were purchased from the central market in Yuhuan City, China. Each weighed 500 ± 50 g, and 160 large yellow croakers were acquired. The large yellow croakers were transported to the laboratory for storage within one hour of purchase. The main reagents of the experiment are shown in Table 1.

### 2.2. Instruments and Equipment

This study utilized a self-developed, tunable, high-voltage electric field device located in Yuhuan, China. The output voltage range of the device was between 1000 V and 9800 V. The strength of the high-voltage electric field was calculated as shown in Equation (1):(1)$$E=\frac{U}{d}$$
where *E* represents the strength of the electric field, measured in kV/m; *U* denotes the device’s output voltage, measured in V; and *d* signifies the electrode spacing, measured in meters.

The high-voltage electric field equipment (Figure 1) was equipped with a safety protection unit that enabled automatic power-off, ensuring safe operation for the staff. Various auxiliary devices, including temperature, humidity, and ozone sensors, monitored the real-time storage environment for optimal preservation. The output end of the high-voltage electric field used electrode strips, which were arranged in a refrigerator (0.7 × 0.4 × 0.7 m) in a configuration of 0.5 × 0.3 × 0.3 m.

### 2.3. Experimental Methods

#### 2.3.1. Sample Preparation

The large yellow croakers were transported to the laboratory with ice cubes to keep them fresh. After arriving at the laboratory, the large yellow croakers were descaled, rinsed with running water, and dried with sterile paper. They were then placed into non-vacuum polyethylene (PE) preservation bags, with each bag containing about 60 cm^3^ of air along with a large yellow croaker. The bags were manually knotted and stored in a refrigerator (Midea, Foshan, China). The equipment, generating high-voltage electric fields, was activated and operated continuously for 21 days.

#### 2.3.2. Preservation Methods

The experiments included four groups, including a control and three exposed to high-voltage electric fields of 5 kV/m, 8 kV/m, and 16 kV/m. Each group contained 40 large yellow croakers. The large yellow croakers were placed in the middle of the electrode strip and stacked together. The electrode strips were located in the middle of the refrigerator. Five samples from each group were collected on days 0, 7, 14, and 21 for testing, during which the physicochemical and microbial indices of the fish were measured and recorded. Then the mean values were then calculated, and the preservation effects of the high-voltage electric fields on the large yellow croakers were analyzed.

#### 2.3.3. Preservation Environment

The storage temperature of large yellow croakers was set to −1 °C (error: ±1 °C), with the humidity value inside the refrigerator maintained at approximately 60.0%. In the experiments, an epitaxial ozone sensor was used to monitor the ozone levels in the refrigerator and inside the polyethylene (PE) preservation bags. A high-voltage electric field device generated ozone, and, due to the refrigerator’s airtightness, the ozone concentration remained stable. The ozone conditions during the storage of large yellow croakers are shown in Table 2.

#### 2.3.4. Sensory Evaluation

Sensory evaluation is a critical indicator of aquatic products’ quality and competitiveness, serving as a decisive factor in consumers’ willingness to purchase and frequency of purchasing aquatic products [23,24]. For large yellow croaker, sensory evaluation is a visual technique of assessing freshness. The evaluation encompasses various sensory characteristics, such as color, odor, mucus, muscle elasticity, eyes, gills, etc.

In this experiment, five professional aquatic product processors provided scores based on the indicators outlined in Table 3. The raters evaluated the large yellow croakers by assessing changes in their sensory indicators. Considering the appearance characteristics, they gave a comprehensive evaluation of their freshness.

#### 2.3.5. pH Value

A 200-watt ordinary blender was used to mix the meat of the large yellow croakers until homogenized. Then, 5 g of the homogenized minced meat was placed in a beaker, followed by the addition of 50 mL of potassium chloride solution. After the mixture had stood for 30 min, the pH value of the sample was measured using a portable pH meter.

#### 2.3.6. Malondialdehyde

The MDA detection method for the large yellow croakers was slightly modified from the high-performance liquid-chromatography method outlined in GB 5009.181-2016 [25]. Firstly, 3 g of minced large yellow croaker meat was weighed and placed in a centrifuge tube. Secondly, 30 mL of trichloroacetic acid mixture was added to the tube, which was then sealed and shaken to mix the contents. Thirdly, the tube was centrifuged at 7500 r/min for 10 min, and the filtrate was extracted for later use.

Derivatization: Firstly, 5 mL of filtrate and standard series solution were each pipetted into separate 25 mL plugged colorimetric tubes. Then, 5 mL of aqueous thiobarbituric acid was added to each tube, which were then plugged to ensure the liquid was mixed well. Next, the tubes were placed in a 90 °C water bath for 30 min. After that, the tubes were taken out and cooled to room temperature. Lastly, an appropriate amount of the supernatant from each tube was taken and filtered through the membrane for separate analysis. The MDA calculation is shown in Equation (2):(2)$$X=\frac{c\times V\times 1000}{m\times 1000}$$
where *X* is the MDA content in the sample, measured in mg/kg; *c* is the concentration of MDA in the sample solution, obtained from the standard series curves, measured in μg/mL; *V* is the constant volume of the sample solution, measured in mL; *m* is the mass of the sample represented by the final sample solution, measured in grams; and 1000 is the conversion factor.

#### 2.3.7. Total Volatile Basic Nitrogen

The TVB-N content in large yellow croakers was measured by the automatic Kjeldahl nitrogen determination method specified in GB 5009.228-2016 [26]. For the test, lean meat from the back of the fish was used. The meat was minced, and 10 g of the sample was weighed and placed into a distillation tube. Next, 75 mL of purified water was added, and the tube was gently shaken to disperse the sample evenly. After allowing the minced meat to macerate for 30 min, 1 g of magnesium oxide was added to the tube and mixed thoroughly. The measurement was then carried out using an 8400 automatic Kjeldahl nitrogen determination instrument (FOSS, Hillerod, Denmark). TVB-N is calculated as shown in Equation (3):(3)$$X=\frac{(V_{1}+V_{2})\times c\times 14}{m}\times 100$$
where *V*_1_ is the titration solution volume consumed by the test, measured in ml; *V*_2_ is the blank titration solution volume, also in ml; *c* is the titration solution concentration, with a unit of 0.1 mol/L; *m* is the mass of the sample, measured in grams; *X* is the basic nitrogen content of volatile salts, expressed in mg/100 g.

#### 2.3.8. Total Viable Counts

The TVC of large yellow croakers was determined based on the method described in GB 4789.2-2016 [27]. The process involved the following steps: First, the large yellow croakers were washed, and their surface moisture was wiped off. Secondly, the back of the fish was sterilized using 75% alcohol cotton, then 25 g of fish meat was weighed and placed in a sterile homogenizing bag. Thirdly, 225 mL of sterile saline was added to the bag, and the contents were homogeneously patted for 60 s. Fourthly, 1 mL of the sample solution was pipetted into a sterile test tube containing 9 mL of diluent solution, and serial 10-fold dilutions were performed up to five times. Fifthly, 1 mL of each dilution was pipetted onto a sterile plate. Once the plate counting agar medium had cooled to 46 °C, the dilution was poured onto the plate and gently swirled to mix the solution well. Lastly, when the plate had cooled, it was inverted and placed in a 30 °C incubator for 72 h.

### 2.4. Statistical Analysis

SPSS Statistics software (version 22) was used to analyze the significance of the differences. The physiological indices of large yellow croaker were examined for their dependence on storage time and electric field strength. Statistical significance was considered at *p* < 0.05.

## 3. Results and Discussion

### 3.1. Sensory Evaluation

As presented in Figure 2(a1–a4), significant differences were observed among the high-voltage electric field groups and the control group of large yellow croakers during storage, in terms of gill brightness, muscle elasticity, volatile odor, and eye appearance. At the same time, the effect of increasing the electric field intensity on these sensory indicators was more significant. Figure 2(a5) shows that the comprehensive sensory score of the large yellow croakers decreased with increased storage time. Between 0 and 7 days, the sensory scores across the four experimental groups remained relatively stable. However, from days 7 to 21, the comprehensive sensory scores for the control group, as well as the 5 kV/m and 8 kV/m high-voltage electric field groups, decreased significantly due to lipid oxidation, protein denaturation, and changes in enzyme activity [28]. This degradation was evident in the softening of the muscle tissue, mucus accumulation on the surface, cloudy eyeballs, and the emission of a decaying odor. In contrast, the sensory evaluation was better maintained over the 21-day storage period when subjected to a 16 kV/m high-voltage electric field, which significantly inhibited spoilage of large yellow croakers.

Figure 3 illustrates the changes in the eye appearance of the large yellow croakers after 21 days of storage under different high-voltage electric fields. The eye turbidity of the large yellow croakers in the control group was significantly higher than that of those in the high-voltage electric field groups. Notably, the eyes of the large yellow croakers remained clear under the influence of a 16 kV/m high-voltage electric field. The results again showed that the higher high-voltage electric field strength had positive effects on the appearance of the large yellow croakers in the range of 0–16 kV/m.

### 3.2. pH Value

Figure 4 depicts the pH variation in the large yellow croakers under distinct electric field strengths. After storage for 7 days, the pH values of the control group and those exposed to 5 kV/m and 8 kV/m high-voltage electric fields were higher than the initial values. Conversely, the pH value of the large yellow croakers subjected to a 16 kV/m high-voltage electric field was lower than the initial value. This reduction in pH was primarily because, at 16 kV/m, the electric field effectively delayed the conversion of carbohydrates (glycogen metabolism to lactate) and the degradation of phosphoric acid by ATPase [21]. Endogenous enzymes in large yellow croakers, such as ATPase and cathepsin, produce small-molecule polypeptides, promote the growth of microorganisms, accelerate the decomposition of proteins, and generate a large number of alkaline substances (e.g., histamine, putrescine), resulting in a gradual increase in pH value. 

The pH values of the large yellow croakers increased from 7 to 21 days in the control group and the groups exposed to 5 kV/m and 8 kV/m high-voltage electric fields. In contrast, the pH value for the 16 kV/m high-voltage electric field group remained lower than that of the control group during the same period. After storage for 21 days, the pH value under the 16 kV/m electric field was the lowest among all conditions. These results indicate that a high-voltage electric field effectively inhibits endogenous enzyme activity in aquatic products, thereby reducing protein decomposition and the production of alkaline substances and slowing the increase in pH value. Additionally, a 16 kV/m electric field may lead to the formation of acidic byproducts, further contributing to the lower pH value.

### 3.3. Malondialdehyde

Figure 5 illustrates the MDA levels in the large yellow croakers over time. In the control group, the MDA levels increased rapidly due to the gradual imbalance in the cellular antioxidant system in the storage environment (−1 °C). From days 7 to 21, the rate of increase in the MDA levels slowed in the high-voltage electric field groups, indicating that the high-voltage electric fields inhibited the lipid oxidation of the large yellow croakers [29,30]. As the intensity of the electric field increased, the rate of lipid oxidation in the large yellow croakers decreased. This effect can be attributed to the higher electric field intensities leading to an enhanced electrical induction phenomenon in the refrigerator [31], reduced free oxidation reactions during storage, and lower levels of primary oxidation products. On the 21st day, the MDA content in the control group and the 5 kV/m, 8 kV/m, and 16 kV/m high-voltage electric field groups were 8.45, 3.55, 3.04, and 2.48 mg/100 kg, respectively. The results indicate that the high-voltage electric fields inhibited lipid oxidation in the large yellow croakers (*p* < 0.05). The high-voltage electric fields effectively reduced the oxidation reaction of the large yellow croakers, which better retained their original color, texture, and flavor.

### 3.4. Total Volatile Basic Nitrogen

Different electric field strengths influenced the freshness of the large yellow croakers during storage at −1 °C, as reflected in the delayed growth trend in the TVB-N levels. Figure 6 shows that the control group maintained at −1 °C had the lowest TVB-N value after 7 days of storage. However, after 14 days, the levels of alkaline substances in the control group and the group subjected to a 5 kV/m high-voltage electric field increased rapidly due to the activity of endogenous enzymes and microorganisms [32]. In contrast, the growth rates of the TVB-N levels in the 8 kV/m and 16 kV/m high-voltage electric field groups were slower. This suggests that the activities of endogenous enzymes and microorganisms are reduced under these higher electric field strengths, leading to attenuated degradation of proteins and nitrogen-containing compounds [33]. At the same time, a high-voltage electric field also mitigates the rate of protein denaturation [34]. It had a positive effect on preserving large yellow croakers. On the 21st day of storage, the TVB-N values of the control group and the groups subjected to high-voltage electric fields of 5 kV/m and 8 kV/m were 48.94 mg/100 g, 41.76 mg/100 g, and 31.32 mg/100 g, respectively, all of which exceeded the edible standard for aquatic products of 30 mg/100 g. However, the TVB-N value for the 16 kV/m high-voltage electric field group was 16.08 mg/100 g, demonstrating a significant preservation effect (*p* < 0.05). A similar preservation effect was observed in tilapia stored at 4 °C under high-voltage fields of 300 kV/m, 600 kV/m, and 900 kV/m, with the TVB-N value being lowest at 900 kV/m after 8 days [31]. These experimental results suggest that high-voltage electric fields can effectively preserve the freshness of large yellow croakers.

### 3.5. Total Viable Counts

The results indicated that the TVC in the control group increased over time, while in the high-voltage electric field groups, the TVC initially decreased and then increased (Figure 7). The primary reason is that the high-voltage electric field suppressed the activity of endogenous enzymes and microorganisms in the large yellow croakers. Additionally, the ozone generated by the high-voltage electric field exerted a bactericidal effect. However, as storage time increased, protein breakdown created favorable conditions for microbial growth. On the 12th day, the TVC in the control group exceeded 5.0 lg CFU/g, surpassing the edible standard. This suggests that although low temperatures inhibit microorganisms in large yellow croakers, the diversity and quantity of microorganisms increase significantly with extended storage, leading to spoilage [35]. The TVCs in the 5 kV/m and 8 kV/m groups exceeded 5.0 lg CFU/g on the 16th day. After 21 days of storage, the TVC was 5.80 lg CFU/g in the control group, 5.60 lg CFU/g in the 5 kV/m group, 5.69 lg CFU/g in the 8 kV/m group, and 4.96 lg CFU/g in the 16 kV/m group. Under the influence of the 16 kV/m high-voltage electric field, the microbial count after 21 days was significantly lower compared to that in the control group, with the average colony counts remaining below 5.0 lg CFU/g, effectively extending their shelf life by 9 days.

Compared to storage at −1 °C, the high-voltage electric field method generates electric ions and penetrates cell membranes, increasing their permeability. This process inactivates microorganisms within the cells. Additionally, a high-voltage electric field can release an electric field corona and ozone in a storage space, which provides sterilization and antibacterial effects [13]. With the continuous enhancement in the intensity of the high-voltage electric field, the ozone content in the refrigerator increased, which had a sterilizing effect on the storage environments, thus delaying the spoilage of the large yellow croakers. 

### 3.6. Preservation Mechanism

Based on the above results, it was observed that during storage, the rapid proliferation of endogenous enzymes and microorganisms contributed to meat spoilage. Microorganisms, which carry amino acid decarboxylase, accelerate protein hydrolysis and enhance the production of biogenic amines and volatile compounds. High-voltage electric field technology aims to inhibit the activity of microorganisms and endogenous enzymes by producing ozone. In addition, the electrical induction generated by the high-voltage electric field reduced the metabolic rate and slowed down lipid oxidation, helping to preserve the original muscle tissue, color, flavor, and nutrients of the large yellow croakers (Figure 8). As a result, the high-voltage electric field effectively extended their shelf life while maintaining the nutrient content.

## 4. Conclusions

In this study, the preservation effects and mechanisms of −1 °C near-freezing storage, combined with three different high-voltage electric fields, on large yellow croakers are investigated. The results indicate that high-voltage electric fields can inhibit the activities of endogenous enzymes and microorganisms and that the stronger the electric field, the more pronounced the effect. The application of high-voltage electric fields slows the growth rate of physicochemical and microbial indices, effectively helping to maintain the fish’s original flavor and nutritional value. The analysis reveals that the main cause of spoilage is due to microbial proliferation during storage stages. The 16 kV/m high-voltage electric field group exhibits the most significant preservation effect. Compared to traditional storage techniques, the high-voltage electric field effectively inhibits spoilage, extends shelf life by 9 days, and increases the market value of large yellow croakers.

The results offer a theoretical foundation for the practical application of high-voltage electric fields in the production, processing, storage, transportation, and marketing of aquatic products. However, the results of this experiment only provide a preliminary reference for the preservation effect of 5–16 kV/m high-voltage electric fields. Future research is needed to explore the use of higher-intensity electric fields to assess the freshness of various types of aquatic products under different conditions. This would help achieve comprehensive freshness preservation throughout the entire chain, from fishing to packaging, transportation, and cold storage.

## Figures and Tables

**Figure 1 foods-13-02877-f001:**
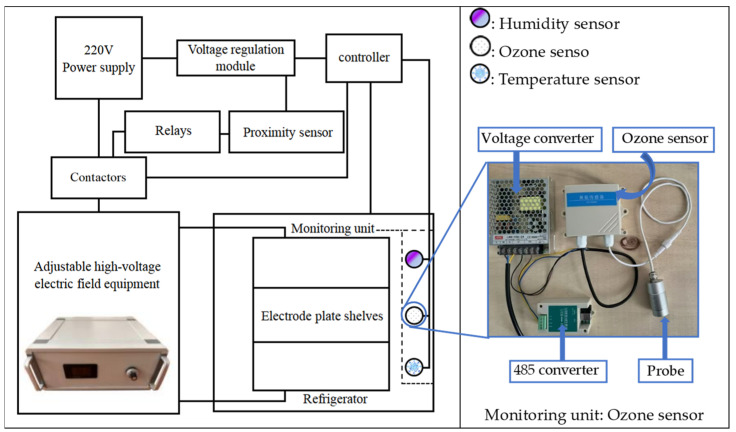
High-voltage electric field generator and monitoring unit.

**Figure 2 foods-13-02877-f002:**
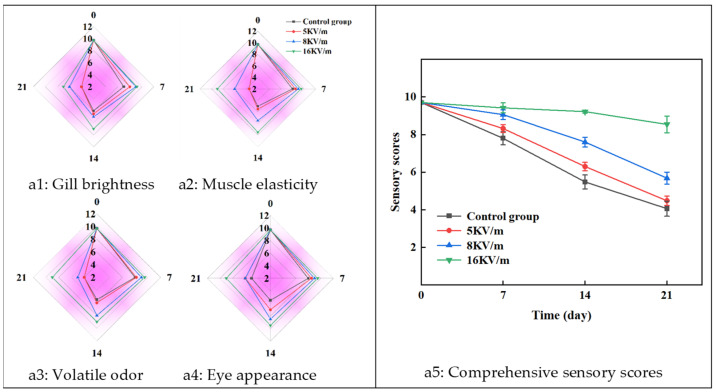
Sensory scores of large yellow croakers subjected to high-voltage electric fields during storage at −1 °C. (**a1**) Gill brightness; (**a2**) muscle elasticity; (**a3**) volatile odor; (**a4**) eye appearance; (**a5**) comprehensive sensory scores.

**Figure 3 foods-13-02877-f003:**
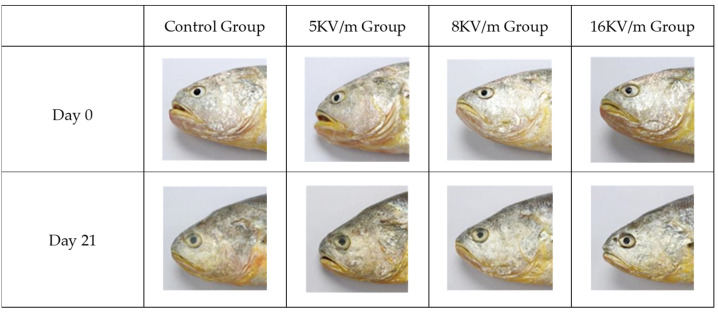
Eye appearance of high-voltage electric fields on large yellow croakers during storage at −1 °C.

**Figure 4 foods-13-02877-f004:**
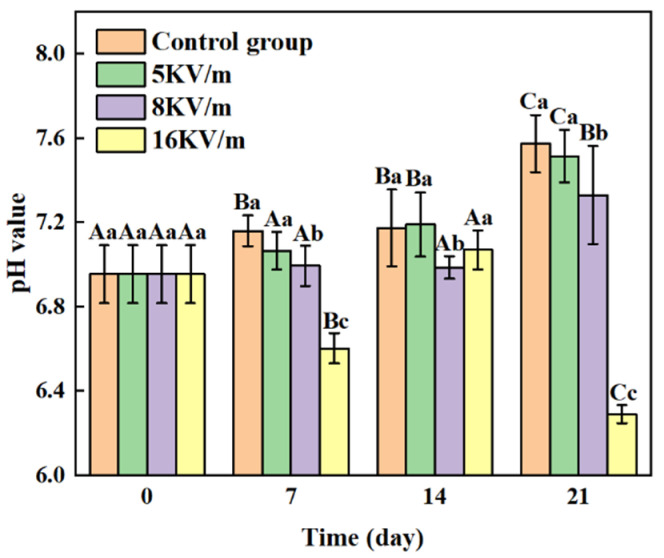
Changes in pH values of large yellow croakers when stored at −1 °C under different high-voltage electric fields. Note 1: Lowercase letters represent significant differences between groups (*p* < 0.05), while capital letters indicate significant differences within the same treatment groups (*p* < 0.05). Data are presented as means ± standard deviations (n = 5).

**Figure 5 foods-13-02877-f005:**
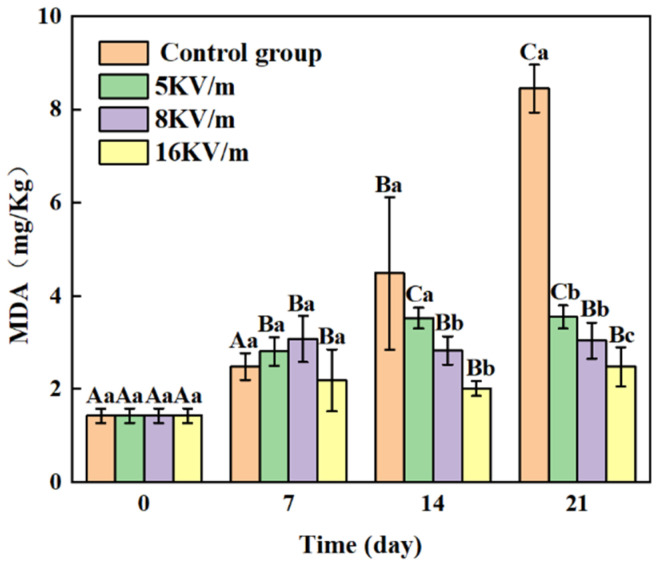
Effects of high-voltage electric fields on MDA levels of large yellow croakers during storage at −1 °C. Note 2: The different letters convey the same meaning as in Note 1. Data are presented as means ± standard deviations (n = 5).

**Figure 6 foods-13-02877-f006:**
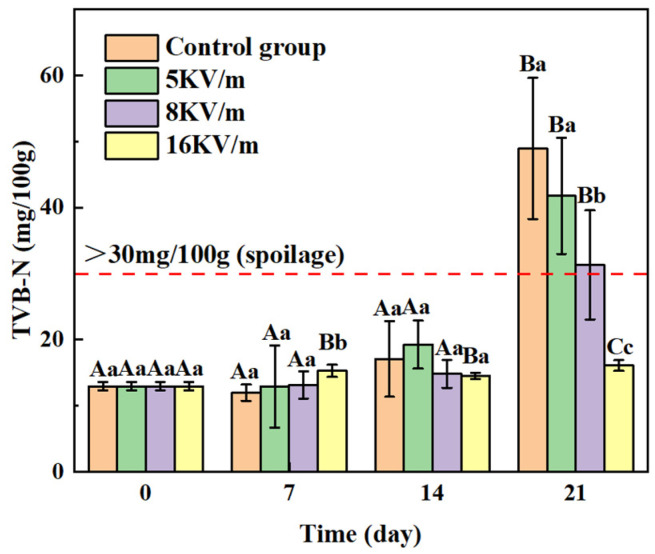
Changes in TVB-N values of large yellow croakers treated with high-voltage electric fields during storage at −1 °C. Note 3: The different letters convey the same meaning as in Note 1. Data are presented as means ± standard deviations (n = 5).

**Figure 7 foods-13-02877-f007:**
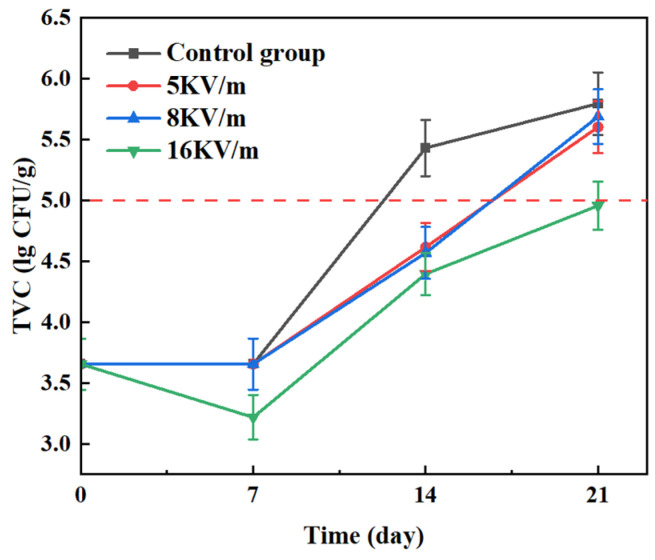
Changes in the TVC of large yellow croakers when stored at −1 °C under high-voltage electric fields.

**Figure 8 foods-13-02877-f008:**
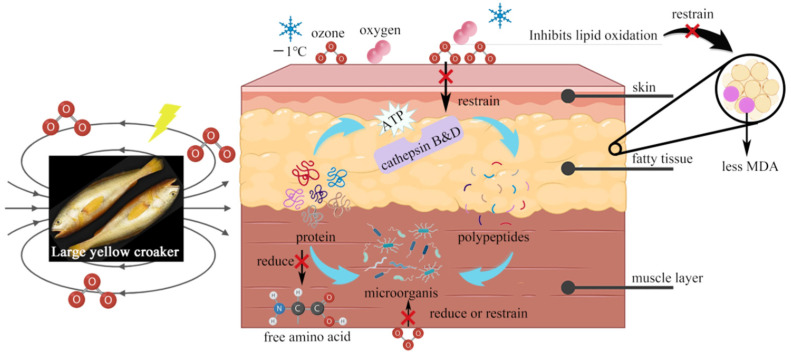
Mechanism of storage and preservation of large yellow croakers using storage at −1 °C combined with high-voltage electric fields.

**Table 1 foods-13-02877-t001:** The main reagents in the experiment.

Test Item	Main Reagents	Manufacturer
pH value	Potassium chloride (KCl)	Sinopharm Chemical Reagent Co., Ltd. (Shanghai, China)
Malondialdehyde	Ammonium acetate	Shanghai Aladdin Biochemical Technology Co., Ltd. (Shanghai, China)
	Trichloroacetic acid Disodium ethylenediaminetetraacetic acid Thiobarbituric acid	Sinopharm Chemical Reagent Co., Ltd. (Shanghai, China)
Total volatile basic nitrogen	Magnesium oxide (MgO) Boric acid (H_3_BO_3_) Hydrochloric acid (HCl) Methyl red indicator (C_15_H_15_N_3_O_2_) Bromocresol green indicator (C_21_H_14_Br_4_O_5_S)	Sinopharm Chemical Reagent Co., Ltd. (Shanghai, China)
	95% ethanol (C_2_H_5_OH)	Hangzhou Qingchen Chemical Reagent Factory. (Hangzhou, China)
Total viable counts	Plate counting agar medium Sterile saline	Guangdong Huankai Microbial Technology Co., Ltd. (Guangzhou, China)

**Table 2 foods-13-02877-t002:** Reference data of the ozone.

Electric Field Strength	In the Refrigerator and Inside PE Preservation Bag Ozone Value (ppm)	Error
control group	0	0
5 kV/m	0.8	±10%
8 kV/m	1.3	±12%
16 kV/m	1.75	±15%

**Table 3 foods-13-02877-t003:** Sensory evaluation indices of large yellow croakers.

Sensory Evaluation Indicators	Grading Principles	Scores
Gill brightness	The gills are bright and mucus-free	8~10
	The gills are darkened and mucus-free	6~8
	The gills are dark red with mucus	4~6
	The gills are black with mucus	2~4
Muscle elasticity	The texture is clear, tightly organized, and elastic	8~10
	The texture is clear, the tissue is soft, and the elasticity is reduced	6~8
	The texture fades, and the tissue is soft and slightly elastic	4~6
	The texture is blurred, the tissue is floating, and it is inelastic	2~4
Volatile odor	No odor	8~10
	Slightly fishy	6~8
	The fishy smell is emanating and slightly putrid smell	4~6
	The whole smells of spoilage	2~4
eye appearance	The eyeball is clear, and the cornea is transparent	8~10
	The eyeball is darkened, and the cornea is clear but slightly sunken	6~8
	The eyeball is darkened, and the cornea is blurred and sunken	4~6
	The eyeball and cornea are blurred as a whole	2~4

Exegesis: sensory score = (gill brightness + muscle elasticity + volatile odor+ eye appearance)/4.

## Data Availability

The original contributions presented in this study are included in this article; further inquiries can be directed to the corresponding author.

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
