# Peer review of "Effect of Near-Freezing Storage Combined with High-Voltage Electric Fields on the Freshness of Large Yellow Croaker"

_foods, 2024, doi:10.3390/foods13182877_

Round 1
Reviewer 1 Report
Comments and Suggestions for Authors
In this study the authors investigated to observe the combine effects of high-voltage electric fields and refrigeration on the comprehensive quality of large yellow croakers, therefore, providing theoretical and data-driven references for the application of cross-regional transportation of fish refrigeration and electric field preservation processes. As overall comment I would like to say that the data and information are greatly useful for further investigation of application of by product for further use. However, there is a lot of rooms for improvement, as well as, better discussed in some issues. Some issues should be addressed before publication. I suggest some minor revision for publication, including below points:
<Design of Experiments>
Line 94 : Please add more information (name, city, etc) of ‘electric field device’.
Line 111 : Please note the full name of PE at the first mention.
Line 113-115 : Please add exposure time of the sample under electric field (ex: all day long for 21 days storages).
Line 127 : Please explain more in details about the how train the sensory panels following which methods and add more information of the panels (sex, age, etc).
Line 134 : Please add more information (name, city, power, etc) of homogenizer.
<Results and discussion>
Please discuss more about main mechanisms of why the results changed with increasing voltages and storage days through whole section of results and discussion.
Line 267 : Please discuss more about main mechanisms of why the TVC decreased with increasing voltages.
<References>
Please double check the references following the author’s guide of the journal.
Author Response
Dear Reviewer,
We would like to thank and acknowledge your constructive comments concerning our manuscript entitled ‘Effect of near-freezing storage combined with high-voltage electric fields on the freshness of large yellow croaker’ (foods-3182801). All the comments have been carefully considered, and modifications marked up using the “Track Changes” function have been made to our manuscript. Below, we address all comments point-by-point in blue.

Reviewer 2 Report
Comments and Suggestions for Authors
This is an interesting study, and valuable in exploring how electric fields can be used to enable use of -1 degree Celsius temperature freezing to prevent spoilage and improve food quality.
The SI unit symbol for kilo should be lower case. For example, 5 KV/m should be 5kV/m. In many places in the paper, KV/m is used and should be replaced by kV/m.
Line 97: “Where” should be lower case: “where”.
Line 110: “rinse” should be “rinsed”.
Line 152: “Where” should be lower case: “where”.
Line 165: “Where” should be lower case: “where”.
Figure 2a: Resolution is poor. It would be good to obtain a higher resolution for the figure.
Figure 7 caption: the negative symbol “-“ for the number -1 degrees C is split on the line before, and looks like a hyphen. The negative sign should be kept with the number. This situation also exists in the Abstract.
Questions/comments on the content:
1. Why were 5, 8, and 16 kV/m selected?
2. It should be clarified how the fish are arranged within the electric field between the electrodes, so that it is understood that the electric field is uniform in a vertical direction. For example is there space on either side of the fish, or are they packed fairly tightly together forming a uniform layer the a 2D cross-section perpendicular to the electric field?
3. The electrode plate shelves in the refrigerator of Figure 1 are separated by a vertical distance d. The paper should explain what ratio of this vertical distance is taken up by the fish, since the dielectric constant of the fish (containing water with high relative permittivity) is higher than that of the air space (relative permittivity =1). Therefore, equation 1 only applies to an empty refrigerator or the case where the space between the electrode plate shelves is fully filled with fish. If the space was not fully filled, then the vertical electric field would not be uniform in the region.
In a partially filled situation for this space, the correct model would be a stacked dielectric situation, which is similar to two capacitors in series (air region and fish region). Therefore, the electric field in each the air region and fish region are different and should be calculated separately, and these calculated values would depend on the ratio of total space within the refrigerator taken up by the fish. The consequences of this situation affect both the electric field corona and ozone production in the air region, and the electric field seen by tissues in the fish. This fact does not change the overall conclusion that higher electric fields help in reducing spoilage. However, it does change the concluded specific electric field value that is needed to reduce spoilage, since the electric field in both the air region and fish region are different and not given by equation 1.
Therefore, if the experiments were conducted for a situation that the space between the electrode plate shelves was not fully filled with fish, the discussion regarding equation 1 should be expanded to calculate the correct electric field in air region and fish region, and based on what ratio of the distance between the electrode plate shelves is taken up by fish, and how much free air space remains. As a secondary comment, from a practical point of view, knowing these separate regional electric fields would enable a stronger conclusion for the paper, and would allow suggestion as to what voltage is needed to be applied between the electrode plate shelves in Figure 1, to create the required air region and fish region electric fields to prevent spoilage. This can be done for different situations, such as when the space between the electrode plate shelves is mostly empty, half full, or completely full of fish. This would greatly strengthen the conclusions of this paper.
Author Response
Dear Reviewer,
We would like to thank and acknowledge your constructive comments concerning our manuscript entitled ‘Effect of near-freezing storage combined with high-voltage electric fields on the freshness of large yellow croaker’ (foods-3182801). All the comments have been carefully considered, and modifications marked up using the “Track Changes” function have been made to our manuscript. Below, we address all comments point-by-point in blue and red.
